# Evaluating community pharmacists' involvement in patient counselling and health education in Nairobi, Kenya: A cross-sectional study

**Seraphine Wanjiro Manjari**⊙*, **Mergia Terefe Ermias**⊙

School of Pharmacy and Health Sciences, Department of Pharmacology and therapeutics, United States International University-Africa, Nairobi, Kenya

* Swmanjari538@gmail.com

## Abstract

### Background

Community pharmacists serve as the most accessible healthcare providers, playing a pivotal role in patient counselling, health education, and disease prevention. Despite their critical function in the healthcare system, limited research has examined their actual involvement in these areas within Nairobi, Kenya. This study aimed to assess community pharmacists' engagement in patient counselling and health education, comparing self-reported practices with observed behaviours using a simulated patient approach.

### Methods

This study employed a cross-sectional descriptive design incorporating both structured questionnaires and a simulated patient (SP) methodology to compare self-reported and observed counselling practices among registered community pharmacies in Nairobi. Sixty pharmacies were randomly selected, and forty-eight pharmacists completed the study. Data were collected using structured questionnaires that assessed pharmacists' self-reported counselling and health education practices, alongside simulated patient visits that evaluated actual pharmacist–patient interactions. The case scenarios included a request for diabetes medication without a prescription, an over-the-counter acne treatment request, and a consultation for severe acne. These encounters were used to assess counselling competence, regulatory adherence, and patient education. Data were analysed using descriptive statistics to compare self-reported practices with observed behaviours.

### Results

While 66.7% of pharmacists reported providing detailed counselling, only 26.7% of simulated visits reflected this practice. Additionally, 75% of pharmacists provided

**Data availability statement:** All relevant data are within the manuscript.

**Funding:** The author(s) received no specific funding for this work.

**Competing interests:** The authors have declared that no competing interests exist.

minimal verbal counselling beyond stating the drug name and dosage. Patient satisfaction levels were low, with most simulated patients rating their encounters poorly on a 20-point scale.

## Conclusion

The study underscores critical gaps in patient counselling, health education, and adherence to prescription regulations among community pharmacists in Nairobi. The disconnect between self-reported and observed counselling practices suggests the need for targeted training programs to enhance pharmacists' communication skills, regulatory compliance, and disease prevention counselling. Strengthening enforcement mechanisms and integrating patient-centred education strategies in pharmacy curricula are essential to optimizing pharmacists' roles in healthcare. These findings highlight the need for targeted training and regulatory enforcement to strengthen pharmacists' roles in patient counselling and health education in Nairobi.

## 1. Introduction

Community pharmacists (CPs) serve as the first point of contact for many patients, offering easily accessible healthcare services without the need for appointments or referrals [1], [2]. Their geographical distribution and relatively lower costs compared to hospital consultations make them an integral part of the healthcare system, particularly in low-resource settings. During dispensing medications, CPs are responsible for ensuring the rational and effective use of drugs by counselling patients on medication adherence, potential side effects, and safety precautions [3]. However, despite their strategic role in patient care, studies indicate that CPs often fail to provide adequate counselling, focusing instead on the transactional aspects of medication dispensing [4], [5].Health education and promotion are essential components of public health, enabling individuals to take charge of their well-being by fostering health-conscious behaviours [6]. Community pharmacists are uniquely positioned to engage in health promotion initiatives, particularly for chronic conditions such as diabetes and hypertension, which are on the rise due to poor lifestyle choices [7]. Through structured educational programs, CPs can provide valuable guidance on medication adherence, lifestyle modifications, and disease prevention, ultimately improving population health outcomes [7]. However, the extent to which CPs are actively involved in these activities remains unclear, particularly in Kenya, where data on their contributions to health education and promotion is scarce.

Traditionally, pharmacists have been viewed primarily as dispensers of medications. The evolving healthcare landscape necessitates a shift toward patient-centred care, where pharmacists actively engage in providing guidance on medication use, disease prevention, and lifestyle modifications [1].Inadequate counselling skills among CPs lead to suboptimal treatment outcomes and medication-related problems [1]. Studies conducted in different settings have demonstrated that pharmacists often fail to adequately counsel patients, resulting in poor medication

adherence and ineffective disease management [8,9]. Despite existing guidelines emphasizing the importance of patient counselling, there remains a gap between the expected and actual practice among CPs.

In Kenya, the Pharmacy and Poisons Board (PPB) regulates community pharmacy practice and provides guidelines that emphasize the pharmacist's responsibility to counsel patients on medication use, safety, and adherence. By law, a licensed pharmacist is expected to be present during pharmacy operating hours, and patient counselling is recognized as an integral component of pharmaceutical care. Furthermore, national policies such as the Pharmacy and Poisons Act and the National Pharmaceutical Policy identify pharmacists as key actors in health promotion and disease prevention, beyond their dispensing role. However, enforcement of these regulations remains inconsistent, and the extent to which pharmacists translate these expectations into routine practice is unclear. Studies from neighbouring countries such as Ethiopia and Uganda similarly highlight gaps, with pharmacists acknowledging the importance of counselling but demonstrating limited engagement in practice [8,10]. These findings underscore the need for country-specific evidence to inform training, regulation, and policy in Kenya.

Several studies have highlighted that, although pharmacists recognize the importance of patient education, their actual involvement in counselling and health promotion is limited [11]. The reasons for this include lack of time, inadequate training, and minimal enforcement of counselling regulations [12]. Inadequate pharmacist-patient interaction can contribute to medication misuse, therapeutic failures, and avoidable adverse effects. Furthermore, in developing countries such as Kenya, limited research has been conducted to assess CPs' engagement in health promotion and disease prevention efforts. Understanding these gaps is essential for informing policy decisions and designing training programs to enhance pharmacists' roles in public health.

Accurate assessment of pharmacist–patient interactions presents methodological challenges. Self-reported surveys may overestimate actual practice due to social desirability bias, while direct observation may alter behaviour. The simulated patient (SP) method offers a valuable solution, as it captures real-world pharmacist behaviour in a standardized manner without disrupting workflow. This approach has been increasingly used in LMICs to evaluate the quality of pharmacy services and is particularly relevant in Kenya, where reliable, practice-based data are scarce.

This study aims to assess the level of involvement of community pharmacists in patient counselling, health education, and health promotion in Nairobi, Kenya. Specifically, the study seeks to evaluate patient counselling practices, determine CPs' engagement in health promotion initiatives, assess the quality of patient education provided, and identify barriers hindering effective counselling and health education. The findings will contribute to bridging the existing gaps in pharmaceutical practice and provide evidence to support regulatory enforcement and policy development aimed at optimizing pharmacists' roles in healthcare.

## 2. Methods

### Study design

This study utilized a cross-sectional descriptive design combining structured questionnaires with a simulated patient (SP) methodology to evaluate the extent of community pharmacists' (CPs) involvement in patient counselling, health education, and health promotion [13]. The study sought to compare self-reported practices of CPs with actual pharmacist-patient interactions observed through simulated patient visits. Across-sectional survey was conducted in parallel with the simulated patient method to provide a comprehensive assessment of pharmacists' engagement in these key aspects of healthcare delivery. Similar designs have been successfully used in pharmacy practice research to capture both self-reported and observed behaviours [8].

### Study setting

The study was conducted across all 17 constituencies in Nairobi, Kenya, between 13th December, 2022 and 28th February, 2023. Community pharmacies included in the study were selected based on their registration status and active licensure

with the Pharmacy and Poisons Board (PPB). The study location was chosen due to the high density of pharmacies and the significant role they play in primary healthcare within urban settings.

## Study population and sampling

**Target population.** The study population comprised all community pharmacies registered with the PPB and holding active operational licenses in Nairobi County.

## Eligibility criteria

**Inclusion criteria.**

- Community pharmacies officially registered with the PPB in Nairobi as at December 2022.
- Pharmacies with valid and active operational licenses at the time of data collection

**Exclusion criteria.**

- Pharmacies that were inactive or not in operation during the study period
- Community pharmacists who declined participation

## Sampling design

**Sampling frame.** According to the Pharmacy and Poisons Board (PPB) records, there were 1,361 registered pharmacies in Nairobi, with 1,213 of them meeting the study's inclusion criteria. The sampling frame consisted of 1,213 registered and operational community pharmacies in Nairobi County. This frame was used to systematically select the study sample, ensuring that the findings would be representative of the broader population of community pharmacies within the county.

## Sampling technique

A stratified random sampling approach was employed to ensure equitable representation across all 17 constituencies in Nairobi. Pharmacies were first categorized into clusters based on their respective constituencies. A proportional random sampling strategy was then applied to select a representative number of pharmacies from each cluster. This method facilitated an unbiased selection process, thereby enhancing the reliability and generalizability of the study findings [13].

## Sample size determination

At the time of the study, there were 1,213 registered community pharmacies in Nairobi. The minimum sample size was calculated using Yamane's formula: $n = N/(1 + N(e)^2)$, where $n$ is the sample size, $N$ is the population size, and $e$ is the margin of error. Using a 95% confidence level and a 10% margin of error, the required sample size was 92 pharmacies. Due to logistical constraints, 60 pharmacies were randomly selected and included in the study. The unit of sampling was the pharmacy, with one pharmacist representative per pharmacy.

## Data collection

## Survey instrumentation and pilot testing

Data collection employed structured questionnaires and simulated patient (SP) scorecards. Both instruments were adapted from previously validated studies in comparable settings [8,11] and were modified to fit the Kenyan community pharmacy context. To ensure clarity and contextual appropriateness, the tools were pretested in six community

pharmacies not included in the final sample. Feedback from this pilot informed refinements such as improving question wording, eliminating redundancies, and adjusting scoring criteria.

Simulated patients participated in a two-day standardized training that included role-play and calibration sessions under faculty supervision. To establish scoring consistency, two independent raters assessed pilot encounters, and inter-rater reliability was confirmed with Cohen's kappa (κ = 0.82), indicating strong agreement and supporting the reliability of the observational scoring process.

The pilot study primarily assessed clarity, feasibility, and inter-rater reliability of the SP checklist. Formal internal consistency testing (e.g., Cronbach's alpha) was not conducted due to the small pilot sample size. Additionally, although the questionnaire was adapted from previously validated instruments, no additional psychometric validation (such as internal consistency analysis) was performed for the survey tool itself. This limitation has been acknowledged in the manuscript.

During the main study, the structured questionnaires were administered to community pharmacists to assess self-reported counselling, health education, and health promotion practices. Immediately after each encounter, SPs completed scorecards evaluating the quality of pharmacist–patient interactions against predefined criteria. This parallel data collection approach enabled comparison of pharmacists' reported practices with their observed behaviours.

## Cross-sectional survey

A structured questionnaire was distributed to the selected community pharmacists. The questionnaire consisted of three main sections: demographic information, patient counselling practices, and pharmacists' involvement in health education and promotion. Responses were measured using a three-point Likert scale (always discussed, discussed sometimes, and never discussed) to assess the extent of pharmacist engagement in these key areas.

## Simulated patient method

To complement the cross-sectional survey, a simulated patient (SP) methodology was employed. Three senior (5th-year) pharmacy students were recruited and underwent a two-day training program involving role-play and calibration sessions supervised by faculty. The training ensured that all SPs could consistently portray the assigned case scenarios and accurately complete evaluation checklists.

The case scenarios focused on two commonly encountered conditions—diabetes mellitus and acne vulgaris—chosen because they require not only pharmacological management but also comprehensive patient counselling, health education, and wellness promotion.

Each of the 60 selected pharmacies was visited once during the study period. Pharmacists were unaware of the specific timing of these visits, thereby reducing the risk of contamination bias or altered behaviour. During each visit, SPs interacted with the pharmacist as genuine clients, following the standardized case scripts.

A structured checklist was used immediately after each encounter to assess key aspects of the pharmacist–patient interaction, including patient history-taking, appropriateness of dispensing, medication counselling, and provision of health education [14]. The checklist parameters mirrored those in the pharmacist-administered questionnaires, enabling direct comparison between self-reported and observed practices.

For quality assurance, two independent raters reviewed all SP-completed checklists. Any discrepancies were discussed and resolved by consensus. This dual coding process strengthened the reliability of the observational data and minimized subjective bias.

## Case scenarios

### Scenario 1: Diabetes medication request without prescription

**Context.** In this scenario, pharmacists were assessed on their ability to recognize that Treviamet® (sitagliptin/metformin) is a prescription-only medicine under the Pharmacy and Poisons Board (PPB) regulations. According to



Kenyan law, community pharmacies are required to operate under the direct supervision of a licensed pharmacist or Pharmaceutical Technologist, who should be present during all operational hours.

### Scenario

Simulated patient 1 visited community pharmacies requesting Treviamet (sitagliptin/metformin) 50/1000 mg without presenting a prescription or offering additional information unless prompted by the pharmacist. Upon further inquiry, the simulated patient provided the following details:

- The medication was intended for personal use.

- They had been diagnosed with diabetes 14 months prior.

- Their last HbA1C test, conducted seven months ago, was 13%.

- A recent fasting blood glucose test (three weeks ago) showed a reading of 9 mmol/L.

- Their last medical review was seven months ago.

    Pharmacists were assessed based on their ability to identify the need for a prescription, provide proper counselling on diabetes management, and offer lifestyle modification advice.

### Scenario 2: Over-the-counter acne treatment request

   **Context.**  In this scenario, pharmacists were assessed on their ability to appropriately recommend and counsel on non-prescription products for the management of mild acne. Under Pharmacy and Poisons Board (PPB) guidelines, pharmacists in Kenya are permitted to supply over-the-counter (OTC) dermatological treatments, provided they ensure patients receive adequate counselling on correct product use, potential side effects, and when to seek further medical advice. Community pharmacies often serve as the first point of care for dermatological conditions in Kenya, given their accessibility and affordability compared to dermatology clinics.

### Scenario

Simulated patient 2 visited community pharmacies requesting benzoyl peroxide and tretinoin for acne treatment. When questioned by the pharmacist, the patient provided the following information:

- The medication was for personal use.

- The patient had inflammatory acne.

- No previous treatments had been attempted.

    Pharmacists were evaluated on their willingness to provide counselling on proper acne treatment, correct medication use, potential side effects, and lifestyle modifications to prevent acne flare-ups.

### Scenario 3: Severe acne consultation

   **Context.**  In this scenario, pharmacists were assessed on their ability to identify severe inflammatory acne as a condition requiring referral to a physician or dermatologist. According to Pharmacy and Poisons Board (PPB) guidelines, pharmacists in Kenya are expected to recognize "red-flag" presentations that cannot be safely managed with over-the-counter products and to refer such patients for specialized medical evaluation. While pharmacists may provide initial advice on supportive care or product use, failure to initiate referral risks treatment delays, complications such as scarring, and inappropriate reliance on suboptimal therapies.



### Scenario

Simulated patient 3 presented at the community pharmacies seeking pharmacist guidance on managing severe inflammatory acne. The patient provided the following details upon further questioning:

- They had been experiencing recurrent acne for the past 24 months.

- Acne flare-ups lasted about two months before subsiding.

- They had previously used tretinoin and benzoyl peroxide but found them ineffective.

- They had only used medications during flare-ups, discontinuing treatment once improvements were noticed or when acne worsened during therapy.

Pharmacists were assessed based on their ability to take a thorough patient history, provide appropriate counselling on acne treatment adherence, and recommend either over-the-counter treatments or referral to a dermatologist if necessary.

Each simulated patient interaction was evaluated using a structured checklist assessing pharmacists' inquiry about patient history, provision of counselling, medication guidance, and health education efforts.

### Ethical considerations

This study adhered to ethical research principles, ensuring participant confidentiality and obtaining informed consent. The study received approval from the United States International University-Africa (USIU-A) Institutional Review Board (IRB) approval number USIU-A/IRB/F068 and the National Commission for Science, Technology, and Innovation (NACOSTI) licence number NACOSTI/P/22/22465. To maintain anonymity, the names and locations of pharmacies were not recorded, and simulated patients were identified numerically.

### Data analysis

Data from structured questionnaires and simulated patient (SP) checklists were entered and cleaned using Microsoft Excel before being analyzed with SPSS version 18. The analysis focused primarily on descriptive statistics, including frequencies, percentages, means, and ranges, to summarize pharmacists' self-reported practices and observed behaviours during SP visits.

Comparisons between self-reported and observed practices were presented descriptively to highlight gaps in counselling, history-taking, and health education. Patient satisfaction scores from SP visits were reported as mean values with ranges to provide an overall impression of interaction quality.

## 3. Results

### Demographic characteristics of participants

A total of 60 community pharmacies were included in the study, with 48 community pharmacists responding to the survey. Among the respondents, 55% (n = 26) held diplomas in pharmacy, while 45% (n = 22) possessed a bachelor's degree in pharmacy. The majority (68%) had over five years of experience in community pharmacy practice, whereas 32%(n = 15) had been in practice for fewer than five years. The study also revealed that 72% (n = 35) of pharmacists worked over 10 hours per day, reflecting the high workload and demanding nature of community pharmacy operations in Nairobi. These findings underscore the potential impact of long working hours on pharmacists' ability to engage in patient counselling and health education effectively.

### Pharmacists' self-reported practices in patient counselling and health education

Survey responses indicated that 71% (n = 34) of pharmacists reported routinely inquiring about patient history, including medication history, symptoms, and allergies, before dispensing medication. Additionally, 66.67% (n = 32) of pharmacists

claimed to provide comprehensive drug information, covering indications, dosage, contraindications, drug interactions, and potential side effects. However, only 50%(n = 24) reported actively engaging in patient education on disease prevention and non-pharmacological treatment approaches, such as lifestyle modifications for chronic conditions like diabetes and hypertension. The findings suggest that while pharmacists acknowledge the importance of patient counselling, its implementation remains inconsistent.

### Findings from simulated patient interactions

**Scenario 1: Diabetes medication request without prescription.** In the diabetes management scenario, where a simulated patient requested Treviamet® (sitagliptin/metformin) without a prescription, only 40% (n = 19) of pharmacists adhered to national regulatory requirements by refusing to dispense the medication without valid documentation. The majority, 60% (n = 29), dispensed the drug either without inquiry or after minimal questioning, despite its classification as a prescription-only medicine under Pharmacy and Poisons Board (PPB) guidelines. This practice raises significant ethical and safety concerns, particularly given the risks of inappropriate dosing and inadequate monitoring in diabetes care. Among those who engaged in further interaction, 35% (n = 17) asked about prior diabetes management and adherence to prescribed therapy, while only 25% (n = 12) provided counselling on lifestyle modifications such as diet and exercise. These findings underscore persistent gaps in both regulatory compliance and the delivery of patient-centred diabetes care within Nairobi's community pharmacies. Similar patterns of non-adherence to prescription regulations have been documented in other low- and middle-income countries, reflecting systemic challenges such as weak regulatory enforcement, commercial pressures, and insufficient pharmacist oversight [15].

### Scenario 2: Over-the-counter acne treatment request

In the acne treatment scenario, 48% (n = 23) of pharmacists correctly identified the appropriate first-line treatment and provided counselling on the use of benzoyl peroxide and tretinoin. However, 36% (n = 17) recommended non-specific analgesics or alternative skincare products without clear justification, and 16% (n = 8) failed to provide structured counselling on acne management. Notably, only 30% (n = 14) of pharmacists discussed additional lifestyle modifications, such as proper skincare hygiene and dietary considerations for acne control. This lack of comprehensive counselling may contribute to suboptimal treatment outcomes for patients managing acne-related concerns.

These findings are consistent with evidence from other low- and middle-income countries, where counselling on OTC dermatological products is often suboptimal due to workload pressures, limited dermatology training in pharmacy curricula, and a transactional orientation toward sales [11,16].In contexts like Kenya, where pharmacies often serve as the first point of care for dermatological conditions, strengthening pharmacists' role in counselling and patient education is critical to ensure safe and effective self-care.

### Scenario 3: Severe acne consultation

For the simulated patient presenting with severe inflammatory acne, 55% (n = 26) of pharmacists recommended continuing the use of over-the-counter treatments, despite the patient reporting poor results with previous medications. Only 20% (n = 10) of pharmacists advised the patient to seek medical consultation with a dermatologist for further evaluation and treatment. Additionally, 45%(n = 22) provided limited counselling, focusing solely on product selection without discussing adherence, potential side effects, or alternative treatment approaches. These findings raise important concerns regarding the clinical management capacity of community pharmacists in Kenya when faced with complex dermatological conditions. Under PPB guidelines, pharmacists are expected to recognize "red-flag" presentations and appropriately refer patients to higher levels of care. Failure to provide timely referral not only risks therapeutic delay but also increases the likelihood of complications such as scarring, antimicrobial resistance from inappropriate antibiotic use, and psychosocial distress. Similar trends have been observed in other low- and middle-income countries, where pharmacists often manage severe

dermatological conditions at the OTC level rather than initiating referral pathways, reflecting both gaps in training and limited enforcement of referral guidelines. Strengthening dermatology-focused content in pharmacy curricula and implementing structured CPD modules may help address these deficiencies and promote safer, more patient-centred care.

### Comparison of self-reported practices and simulated patient observations

A significant discrepancy was observed between pharmacists' self-reported practices and their actual counselling behaviour during simulated patient visits. While 71% (n = 34) of pharmacists claimed to routinely inquire about patient history, only 15.26% (n = 7) actively engaged in a thorough patient assessment when observed in practice. Similarly, although 66.67% (n = 32) of pharmacists reported providing comprehensive drug information, only 26.67% (n = 13) of simulated patients received adequate guidance regarding medication use. These inconsistencies highlight a gap between perceived and actual pharmacist performance in patient counselling and health education.

### Patient satisfaction and medication labelling

Simulated patients assessed their interactions with pharmacists based on communication clarity, counselling effectiveness, and overall satisfaction. The results showed that 59% (n = 28) simulated patients rated their pharmacist encounters between 0–5 on a 20-point scale, indicating a generally low level of satisfaction. Additionally, 75% (n = 36) of pharmacists failed to provide verbal counselling beyond stating the drug name and dosage, while 64% (n = 31) of prescriptions lacked critical labelling details, including the patient's name, dosage schedule, and administration instructions. The lack of comprehensive labelling and verbal counselling poses a potential risk to medication safety and adherence.

## 4. Discussion

The findings of this study highlight significant discrepancies between community pharmacists' self-reported counselling practices and their actual observed behaviour during simulated patient visits. While pharmacists generally acknowledged the importance of patient counselling and health education, the study revealed notable inconsistencies in their implementation. These results suggest the need for targeted interventions to enhance pharmacists' engagement in patient-centred care.

### Discrepancies between self-reported and observed practices

A key finding of this study was the substantial gap between pharmacists' self-reported practices and their observed behaviour during simulated patient (SP) visits. While 71% (n = 34) of pharmacists indicated that they routinely asked about patient history, only 15.3% (n = 7) were actually observed doing so. Similarly, although 66.7% (n = 32) reported providing comprehensive drug information, only 26.7% (n = 13) of SP encounters reflected adequate counselling. This mismatch between perception and practice highlights a persistent disconnect that has been noted in other low- and middle-income countries (LMICs), including Ethiopia, Nigeria, and Saudi Arabia, where pharmacists often over-report counselling activities compared to what is observed in practice.

Several factors may explain these discrepancies. High patient loads and commercial pressures in community pharmacies may limit the time pharmacists can dedicate to detailed counselling. Inadequate training in communication skills and lack of reinforcement of counselling standards during pharmacy education also contribute to limited patient engagement. Furthermore, weak enforcement of existing regulatory frameworks in Kenya, such as those from the Pharmacy and Poisons Board, may reduce the incentive for pharmacists to consistently provide patient-centred care.

### Regulatory adherence and ethical considerations in medication dispensing

One of the most concerning findings was the frequent non-adherence to prescription regulations. In the diabetes management scenario, 60% (n = 29) of pharmacists dispensed Treviamet without a valid prescription, despite its clear

classification as a prescription-only medicine. Such practices raise serious ethical and regulatory concerns, as unsupervised use of antidiabetic medicines can result in inappropriate dosing, dangerous drug interactions, poor glycaemic control, and heightened risk of complications such as hypoglycaemia.

This behaviour is not unique to Kenya; "Similar findings have been reported in Ethiopia, where [8] observed a marked gap between self-reported counselling and observed counselling practices among community pharmacists. Non-prescription dispensing is also common in Kenya and other Sub-Saharan African countries [8,9,17].These patterns point to systemic issues, including weak regulatory enforcement, economic dependence on medicine sales, and insufficient accountability mechanisms in community pharmacy practice. Moreover, inadequate training on ethical standards and the absence of consistent inspection frameworks may create an environment where regulatory non-compliance becomes normalized.

Despite many pharmacists acknowledging the importance of counselling and education, simulated patient visits demonstrated inconsistencies in history-taking, medication counselling, and adherence to prescription regulations. One possible contributing factor is the lack of dedicated consultation spaces in many community pharmacies in Nairobi, which may limit opportunities for confidential and comprehensive counselling. Similar infrastructural constraints have been reported in other low- and middle-income countries, where the absence of private counselling areas has been identified as a barrier to effective patient education and pharmacist–patient communication [11].

### Limitations in disease prevention and non-pharmacological counselling

The study also highlighted gaps in pharmacists' engagement in preventive health counselling and non-pharmacological management. While half of the pharmacists reported providing education on disease prevention, lifestyle modifications, and non-pharmacological interventions, only 30% of simulated patient encounters had such advice. This lack of engagement is particularly concerning in the management of chronic diseases such as diabetes and hypertension, where lifestyle modifications play a crucial role in treatment outcomes. Additionally, the limited discussion of lifestyle factors in acne management suggests a need for improved pharmacist training in dermatological conditions and patient education strategies. Although these findings show an improvement those found by [18] where none of the pharmacists involved in the study educated simulated patients on drug interactions, adverse reaction and any non-pharmacological alternatives there is more that can be done.

### Pharmacists' role in dermatological counselling

The findings related to acne management further reinforce the need for enhanced pharmacist training. Although nearly half (48%) of pharmacists correctly identified appropriate first-line treatments for acne, a substantial proportion (36%) recommended non-specific analgesics or alternative skincare products without clear justification. Furthermore, only 20% of pharmacists recommended referral to a dermatologist for patients presenting with severe acne. This indicates a lack of confidence or knowledge regarding appropriate referral pathways and comprehensive dermatological counselling, underscoring the necessity of continued professional development in this area.

### Patient satisfaction and medication labelling

Simulated patients assessed their encounters with pharmacists using a structured 0–20 satisfaction scale adapted from [8], which evaluates communication clarity, counselling effectiveness, and overall satisfaction. This scale has been previously validated in pharmacy practice research and was selected for its ability to capture multiple dimensions of patient–pharmacist interaction in a simple numeric format. To ensure reliability, simulated patient scorecards were independently reviewed by two faculty supervisors, and discrepancies were resolved by consensus.

In this study, satisfaction scores ranged from 0 to 15, with a mean of approximately 6 out of 20, reflecting generally low levels of satisfaction. Nearly 59% (n = 28) of encounters were rated between 0–5, indicating poor engagement and limited communication. In addition, 75% (n = 36) of pharmacists provided no counselling beyond stating the drug name and

dosage, while 64% (n = 31) of dispensed medicines lacked critical labelling details such as patient name, dosage schedule, and administration instructions. These findings underscore the risks of poor counselling and inadequate labelling for medication safety and adherence.

## Implications for policy and practice

The findings of this study have significant implications for pharmacy practice, regulatory bodies, and pharmacy education. There is an urgent need for continued professional development programs to enhance pharmacists' competency in patient counselling, particularly in chronic disease management, dermatological conditions, and lifestyle modification strategies. Regulatory agencies must also strengthen enforcement measures to ensure adherence to prescription guidelines and improve medication labelling standards. Additionally, pharmacy education curricula should place greater emphasis on communication skills and patient-centred care approaches to better prepare future pharmacists for their roles in healthcare delivery.

## Limitations

While this study provides novel insights into community pharmacy practice in Nairobi, certain boundaries should be acknowledged. First, this study relied solely on descriptive statistics, which limited the ability to examine associations between pharmacist characteristics and counselling practices. Future studies with larger datasets should incorporate inferential analyses to explore these relationships more comprehensively. Second, although unannounced simulated patient (SP) visits were used to minimize bias, the possibility of subtle behaviour modification (Hawthorne effect) cannot be completely ruled out. Third, we did not include qualitative triangulation, which could have enriched understanding of the contextual drivers behind observed practices. Fourth, the sample focused on Nairobi pharmacies, which offers valuable urban insights but may not fully represent rural or regional variations. Fifth, the achieved sample size (48 pharmacists) was lower than the calculated minimum required sample of 92, primarily due to logistical challenges. This reduced sample size may limit the generalizability of the findings. Sixth, although the tools were adapted from validated instruments and pilot tested, formal psychometric validation—such as internal consistency testing (e.g., Cronbach's alpha)—was not performed, which may limit the overall measurement robustness. Finally, although SPs underwent structured training and inter-rater reliability was acceptable, minor subjective variation in scoring may have remained.

## Recommendations for future research

Further research is needed to explore the barriers preventing pharmacists from fully engaging in patient counselling and education. Qualitative studies could provide deeper insights into pharmacists' perceptions of their roles, workload challenges, and potential strategies for improving counselling practices. Additionally, intervention studies evaluating the effectiveness of targeted training programs in enhancing pharmacists' counselling skills and regulatory compliance would be valuable in guiding future practice improvements.

## 5. Conclusion

This study highlights the crucial role of community pharmacists in patient counselling, health education, and medication safety. While pharmacists serve as frontline healthcare providers, the findings reveal significant gaps between their self-reported practices and actual interactions with patients. Despite many pharmacists acknowledging the importance of counselling and education, simulated patient visits demonstrated inconsistencies in history-taking, medication counselling, and adherence to prescription regulations.

The lack of comprehensive patient assessments and inadequate provision of medication-related information raise concerns about patient safety and therapeutic outcomes. Additionally, deficiencies in medication labelling and verbal

counselling further contribute to the risk of medication errors and poor adherence. These findings underscore the need for structured interventions aimed at improving pharmacists' engagement in patient-centred care.

To address these gaps, targeted pharmacist training programs should be implemented to enhance communication skills, regulatory compliance, and disease prevention counselling. Strengthening enforcement mechanisms for prescription regulations is equally critical to ensure adherence to ethical dispensing practices. Furthermore, pharmacy education curricula should place greater emphasis on practical counselling techniques and real-world patient interactions.

Improving pharmacists' involvement in health education and disease prevention is essential for optimizing healthcare delivery in Nairobi and beyond. Future research should explore the barriers that hinder pharmacists from fully engaging in patient counselling and evaluate the effectiveness of training interventions designed to bridge these gaps. By prioritizing pharmacist training, policy enforcement, and patient-centred education strategies, community pharmacies can significantly contribute to improved medication safety and public health outcomes.

## Acknowledgments

We would like to appreciate the willingness of supervisors, community pharmacists and simulated patients for their vital role in this study.

## Author contributions

**Conceptualization:** Seraphine Wanjiro Manjari, Mergia Terefe Ermias.

**Data curation:** Seraphine Wanjiro Manjari.

**Formal analysis:** Seraphine Wanjiro Manjari.

**Investigation:** Seraphine Wanjiro Manjari.

**Methodology:** Seraphine Wanjiro Manjari, Mergia Terefe Ermias.

**Supervision:** Mergia Terefe Ermias.

**Validation:** Mergia Terefe Ermias.

**Writing – original draft:** Mergia Terefe Ermias.

**Writing – review & editing:** Seraphine Wanjiro Manjari.

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
