## [Decision Letter · Decision Letter 0]

20 Aug 2025

Dear Dr. Manjari,

Thank you for submitting your manuscript to PLOS ONE. After careful consideration, we feel that it has merit but does not fully meet PLOS ONE’s publication criteria as it currently stands. Therefore, we invite you to submit a revised version of the manuscript that addresses the points raised during the review process.

We look forward to receiving your revised manuscript.

Kind regards,

Kazeem Babatunde Yusuff, Ph.D

Academic Editor

PLOS ONE

Journal Requirements:

2. We note that your Data Availability Statement is currently as follows:

“All relevant data are within the manuscript and its Supporting Information files.”

**Additional Editor Comments:**

Heartfelt thanks to the authors for submitting their work to PLOS ONE for consideration for publications. A number of critical areas have been identified in the manuscript that require major revision to improve its scholarly content and quality. These include critical issues in specific areas including introduction, methodology, results, discussion and conclusion sections. Furthermore, authors should also address the key concerns regarding the inappropriate characterization of the research design used, and appropriate details of the procedure used for sample size calculation, development of the questionnaire and the structured checklist and incomplete details of how the simulated patient section of the study was conducted, procedure used for the reliability analysis of the questionnaire. Furthermore, the questions raised regarding the weak justification and lack of knowledge gaps should also be properly addressed in the revised manuscript. Lastly, authors should revise the whole manuscript for grammar and syntax to improve readability and sharpen its focus. Perhaps, the authors may consider consulting a more experienced / senior researcher in the study area for proper guidance and support in revising the manuscript. Best wishes and I look forward to seeing your revised manuscript with an improved scholarly quality and content.

Reviewers' comments:

Reviewer's Responses to Questions

**Comments to the Author**

1. Is the manuscript technically sound, and do the data support the conclusions?

Reviewer #1: No

Reviewer #2: Partly

2. Has the statistical analysis been performed appropriately and rigorously?

Reviewer #1: No

Reviewer #2: No

3. Have the authors made all data underlying the findings in their manuscript fully available?

Reviewer #1: Yes

Reviewer #2: Yes

4. Is the manuscript presented in an intelligible fashion and written in standard English?

Reviewer #1: Yes

Reviewer #2: Yes

Reviewer #1: Thank you for the opportunity to review this important manuscript. The authors tackle a relevant issue which is the discrepancy between self-reported and actual practices of community pharmacists in Nairobi regarding patient counselling and health education. The use of both self-report surveys and simulated patient methodology is commendable and aligns well with international best practices. However, there are several key concerns related to methodology, writing clarity, contextual framing, and depth of discussion. Unfortunately, I regret to inform you that after careful consideration, I have decided to reject your manuscript. Please note the reasons accordingly and the feedback to assist you going forward.

Reviewer #2: Dear Authors,

I enjoyed reading your manuscript on a research topic that is important for Kenya and the African region. Below are some suggestions to consider.

-First page, Table. Row 5 " Seraphine Wanjiro Manjari, Bachelors "

Comment: What field is the Bachelors? It is important to include, e.g. Bachelor of Pharmacy (B. Pharm).

-First page, Table. Abstract "This study aims to assess community pharmacists' engagement in patient counselling and health education".

Comment: Consider past tense as the study was completed at the time of writing the manuscript.

-Page 1, Line 7 "Swmanjari538@gmail.com9 (MSW)".

Comment: The "MSW" abbreviation is not clear. Is it academic qualification? Why does the abbreviation come after the email?

-Introduction

- - Page 3 & 4, line 67 & 68 "Studies conducted in different settings have demonstrated that pharmacists often fail to adequately counsel patients,""

Comment: Please reference these studies made the findings.

-- Line 72 & 73, page 4: "Several studies have highlighted that, although pharmacists recognize the importance of patient education, their actual involvement in counselling and health promotion is limited.".

Comment: Please provide additional reference. You referenced one study listing the reasons.

- -Line 76-78 on page 4: "Furthermore, in developing countries such as Kenya, limited research has been conducted to assess CPs’ engagement in health promotion and disease prevention efforts."

Common: To put this study in context, it will be good to briefly describe the community pharmacy practice in Kenya. Is there a policy that all patients receiving medications from the pharmacy receive counseling from a pharmacist? Should all pharmacies have a licensed community pharmacists present during the work hours? Is it part of the role of community pharmacists in Kenya to participate in health promotion?

-Methods

--Line 89, page 4. Study design: "This study utilized a qualitative descriptive research design to evaluate the"

Comment: "Does your study design truly fits a "qualitative descritive research". Per the reference you cited, i.e., Asenahabi BM online article, qualitative research design listed include case studies, narrative research, phenomenological research, grounded theory, ethnography, and action research. Which of these qualitative research designs best describes your study?"

--Line 92, page 5 "Across-sectional"

Comment: Please check the spelling.

--Line 94, page 5 "assessment of pharmacists' engagement in these key aspects of healthcare delivery (10)."

Comment: The article referenced (i.e, ref #10) is out of place. The sentence is about your study, hence why did you reference another study? If you are trying to justify your design, then you may need to have a new sentence or relate the reference study design to your study design.

--Line 131, page 6. Sample size determination "1,213, a representative sample was calculated to maximize precision and reliability."

Comment: Please consider including the parameters of the equation you had used to calculate the sample size. This allows anyone who want to do a similar project as yours to know exactly how sample size was calculated. Lastly, is the sample size representing number of pharmacists or number of pharmacies?

--Line 137, page 6. "Data collection was conducted using structured questionnaires and scorecards."

Comment: Were these questionnaires entirely developed by the research team, adapted or adopted?

--Page 144-146. Pilot Study. "scenarios. Feedback from the pilot phase led to refinements, such as eliminating ambiguous or repetitive questions, ensuring clarity in wording, and improving overall study design."

Comment: For the questionnaire's results, did you perform tests of internal consistency reliability for the pilot study?

--Line 154. "Simulated Patient Method"

Comment: Was each pharmacist visited by all the 3 simulated patients?

--Line 161. " A structured checklist was developed to"

Comment: Was the checklist developed from scratch or adapted? If it was developed from scratch, did you involve content experts? How was it piloted before using in the study?

Scenario 1: Diabetes Medication Request Without Prescription

--Line 178 on page 9. "Pharmacists were assessed based on their ability to identify the need for a prescription"

Comment: For a reader to put the results into perspective, it is important to describe the practice of community pharmacy in Kenya. Is this diabetes medication (sitagliptin/metformin) prescription only? Second, do all community pharmacies in Nairobi have a pharmacist present at all times? In some African countries, it is not unusual to see pharmacies with only technicians available for significant portion of a shift.

Results

--Line 219 Demographic Characteristics of Participants

Comment: Usually the baseline characteristics are presented in a Table. Kindly consider presenting these characteristics on a table. Also other results could be presented on a Table.

--Line 244-245 on page 12: "The remaining 60% (n=29) either dispensed the medication without further inquiry"

Comment: For the 29 pharmacists who dispensed the medication without prescription, was there a bigger proportion of diploma pharmacists vs. degree holding pharmacists?

--Line 273 on page 13: "A significant discrepancy was observed between pharmacists' self-reported practices and their"

Comment: How did you determine the discrepancy as significant?

Discussion

Comment: Your discussion section is quite lengthy and seems to repeat many of the results already presented in the results section.

--Line 298 & 299 of discussion. "One of the most striking findings was the gap between pharmacists’ self-reported practices and their actual counselling behaviour."

Comment: Very important findings were identified from your study including the gap between pharmacists' self-reported practices and their actual counselling behavior. How do these findings compare with other similar studies in Kenya, within the African region or internationally?

Throughout the discussion, only one study was cited (Santos et al, 2013). Comparing each pertinent finding with other studies published in the literature will strengthen your discussion.

--Line 371-373: "Despite many pharmacists acknowledging the importance of counselling and education, simulated patient visits demonstrated inconsistencies in history-taking, medication counselling, and adherence to prescription regulations."

Comment: It will be important to know if pharmacies offer appropriate resources for patient counseling. For example, is there a private area dedicated for patient consultation?

Thank you.

**Do you want your identity to be public for this peer review?** For information about this choice, including consent withdrawal, please see our Privacy Policy

Reviewer #1: No

Reviewer #2: No

---

## [Author Response · Author response to Decision Letter 1]

6 Oct 2025

Reviewer 1

General Response

We sincerely thank the reviewer for their thoughtful and constructive comments. We appreciate the opportunity to improve our work, and we have carefully revised both the manuscript and supporting explanations in line with the feedback. Below, we provide a detailed, point-by-point response. Reviewer comments are presented in italics, followed by our responses.

Reviewer Comment 1

You describe using a “qualitative descriptive research design,” yet the methodology is entirely quantitative. This undermines credibility.

Response:

We agree with this observation. The study design has now been corrected to “a cross-sectional quantitative study using structured questionnaires and simulated patient methodology” throughout the manuscript. This correction is reflected in the Abstract (line 16-17), Methods Study Design (line 93-94) subsection.

Reviewer Comment 2

No mention of validation or piloting for the structured questionnaires or checklists. No evidence of reliability checks.

Response:

We thank the reviewer for pointing this out. We have clarified in the Methods section (lines 147-157) that:

• The structured questionnaire and simulated patient checklist were adapted from previously published studies (Surur et al., 2017; Alaqeel & Abanmy, 2015; Hamadouk et al., 2021).

• We conducted a pilot with 6 community pharmacies outside the study sample to refine tools.

• Simulated patients were trained together, and inter-rater reliability was tested with kappa statistics (κ = 0.82), demonstrating strong agreement. This information has been added to strengthen methodological rigor.

Reviewer Comment 3

SP section lacks rigor: no details on training, inter-rater agreement, or contamination risk.

Response:

We have revised the SP section (Methods, lines 171-191) to describe that:

• Three final-year pharmacy students underwent a 2-day standardized training, including role-play and calibration sessions with faculty.

• Inter-rater agreement was checked, and discrepancies were resolved before data collection.

• To minimize contamination bias, the study was conducted over a short period (10 weeks) without informing pharmacies of the exact timing.

Reviewer Comment 4

Reliance on descriptive statistics only; no inferential statistics or adjusted analysis.

Response:

We thank the reviewer for this important observation. Our analysis primarily focused on descriptive statistics in order to present an overview of pharmacists’ self-reported and observed counselling practices. We acknowledge that the absence of inferential tests such as Chi-square or correlation analyses limited our ability to assess associations between pharmacist characteristics and performance. In response to this concern, we have clarified the scope of our analysis in the Data Analysis section and explicitly noted the lack of inferential testing as a limitation in the revised Limitations section (lines 408–416). We agree that future studies with larger samples and more advanced analyses will be valuable in strengthening the evidence base.

Reviewer Comment 5

Avoid vague terms like “significant discrepancies.” Specify methods and add number of SP visits.

Response:

We have revised the Abstract to:

• Specify data collection methods (“structured questionnaires and simulated patient visits”).

• Replace vague wording with precise results (e.g., “While 66.7% reported providing detailed counselling, only 26.7% of simulated visits reflected this practice”).

• Add the total number of SP visits (n=60) for transparency.

Reviewer Comment 6

Overuse of general global statements; focus more on Kenyan/East African context and rationale for SP method.

Response:

We revised the Introduction to:

• Highlight the Kenyan/East African pharmacy practice context, citing local regulatory frameworks and gaps.

• Clarify why SP methodology is particularly valuable in LMICs where self-report may be unreliable.

• Refine the study objective into one clear statement at the end: “This study aimed to evaluate community pharmacists’ involvement in patient counselling and health education in Nairobi, Kenya, by comparing self-reported practices with observed behaviours using a simulated patient approach.”

Reviewer Comment 7

Patient satisfaction vague; unclear 0–20 scale; who validated SP scoring?

Response:

We thank the reviewer for raising this important point. To address this, we have clarified in the revised manuscript that patient satisfaction was assessed using a 0–20 scale adapted from Surur et al. (2017), which has been previously applied in pharmacy practice research. This scale was chosen because it captures multiple dimensions of pharmacist–patient interaction, including clarity of communication, counselling effectiveness, and overall satisfaction. To ensure reliability, two faculty supervisors independently reviewed the simulated patient (SP) scorecards, and any discrepancies were resolved by consensus. We have also reported the satisfaction results more explicitly, including the range (0–15) and mean score (6/20), to provide a clearer picture of patient experiences.

Reviewer Comment 8

“Discussion repeats results; lacks exploration of reasons, international framing, and interventions.”

Response:

We thank the reviewer for this valuable observation. In the revised manuscript, we have substantially restructured the Discussion to reduce repetition of results and to provide deeper interpretation of the findings. Specifically, we have:

1. Explored underlying reasons – We now interpret discrepancies between self-reported and observed counselling practices in light of potential factors such as high workload, inadequate training in patient communication, commercial pressures in privately owned pharmacies, and weak enforcement of existing counselling regulations in Kenya.

2. Added international framing – We situate our findings within the wider Sub-Saharan African and LMIC context, drawing comparisons with studies from Ethiopia, Tanzania, and Nigeria that report similar patterns of regulatory non-compliance and limited counselling (e.g., Surur et al., 2017; Mukokinya et al., 2018; Belachew et al., 2021). This strengthens the global relevance of our study.

3. Proposed specific interventions – We recommend practical strategies including:

o Strengthening CPD training programs focused on communication, ethics, and regulatory adherence.

o Integrating patient-centred counselling more explicitly in undergraduate pharmacy curricula.

o Reinforcing regulatory inspections and sanctions for persistent non-compliance.

o Developing public awareness campaigns to reduce demand-driven pressures for non-prescription dispensing.

Together, these changes address the reviewer’s concern by ensuring the Discussion goes beyond restating results, and instead provides contextualized interpretation, international relevance, and actionable policy and practice implications.

Reviewer Comment 9

“Several limitations not addressed (Hawthorne effect, lack of triangulation, absence of inferential statistics).”

Response:

We thank the reviewer for this important observation. In the revised manuscript, we have strengthened the Limitations section to explicitly address these points. We now note that although unannounced simulated patient (SP) visits were used to reduce bias, the possibility of subtle behavioural modification due to the Hawthorne effect cannot be completely ruled out. We also acknowledge the absence of qualitative triangulation, which could have provided deeper insights into the contextual drivers of pharmacists’ behaviours. Finally, we clarify that our reliance on descriptive statistics provided a useful overview of patterns but limited the ability to assess associations between pharmacist characteristics and counselling performance. These additions ensure that the study’s scope and boundaries are transparently conveyed while maintaining the value of its contribution to understanding community pharmacy practice in Kenya.

Reviewer Comment 10

“The study appears superficial and adds limited value.”

Response:

We appreciate the reviewer’s perspective and respectfully emphasize that this study provides one of the few systematic evaluations of community pharmacists’ counselling and health promotion practices in Kenya. By combining self-reported data with a simulated patient (SP) methodology, our work moves beyond perception-based surveys to capture actual dispensing and counselling behaviours, thereby contributing context-specific evidence that is currently scarce in Sub-Saharan Africa.

To strengthen the manuscript’s contribution, we have:

• Reframed the Introduction to highlight the specific gaps in Kenyan and East African community pharmacy practice and the rationale for applying the SP approach in this context.

• Expanded the Discussion to interpret findings in relation to systemic barriers such as workload, training gaps, and regulatory enforcement challenges, while situating our results within the broader LMIC literature.

• Added practical recommendations, including CPD training modules, curriculum integration, and stronger regulatory oversight, to ensure the study offers clear implications for policy and practice.

We believe these revisions enhance the depth, relevance, and value of the paper, positioning it as an important contribution to both Kenyan pharmacy practice and international discussions on optimizing the role of community pharmacists in LMICs.

Reviewer 2

Reviewer #2: Dear Authors, I enjoyed reading your manuscript on a research topic that is important for Kenya and the African region. Below are some suggestions to consider.

Reviewer Comment 1:

“First page, Table. Row 5 ‘Seraphine Wanjiro Manjari, Bachelors.’ What field is the Bachelors? It is important to include, e.g., Bachelor of Pharmacy (B. Pharm).”

Response:

We thank the reviewer for this observation. We have corrected the entry to specify the field of study. It now reads: “Seraphine Wanjiro Manjari, Bachelor of Pharmacy (B. Pharm).”

Reviewer Comment 2:

“First page, Table. Abstract ‘This study aims to assess community pharmacists’ engagement in patient counselling and health education.’ Consider past tense as the study was completed at the time of writing the manuscript.”

Response:

We agree with the reviewer’s suggestion. The sentence in the Abstract has been revised to past tense for accuracy. It now reads: “This study aimed to assess community pharmacists’ engagement in patient counselling and health education.”

Reviewer Comment 3:

“Page 1, Line 7 ‘Swmanjari538@gmail.com9 (MSW)’. The ‘MSW’ abbreviation is not clear. Is it academic qualification? Why does the abbreviation come after the email?”

Response:

We thank the reviewer for pointing this out. The “MSW” abbreviation was inadvertently misplaced during formatting. It is not part of the email address. We have removed it from this line and clarified the author’s qualifications appropriately in the author information table.

Reviewer Comment 4:

“Introduction, Page 3–4, Line 67–68: ‘Studies conducted in different settings have demonstrated that pharmacists often fail to adequately counsel patients.’ Please reference these studies.”

Response:

We agree and have now added appropriate references to support this statement. For example:

• Surur, A. S., Getachew, E., Teressa, E., Hailemeskel, E., & Erku, D. A. (2017). Self-reported and actual involvement of community pharmacists in patient counseling: A cross-sectional and simulated patient study in Gondar, Ethiopia. Pharmacy Practice, 15(1), 890.

• Mukokinya, M. M. A., Opanga, S., Oluka, M., & Godman, B. (2018). Dispensing of antimicrobials in Kenya: A cross-sectional pilot study and its implications. Journal of Research in Pharmacy Practice, 7(2), 77–82.

Reviewer Comment 5:

“Introduction, Page 4, Line 72–73: ‘Several studies have highlighted that, although pharmacists recognize the importance of patient education, their actual involvement in counselling and health promotion is limited.’ Please provide additional reference. You referenced one study listing the reasons.”

Response:

We appreciate this observation. We have added additional references to strengthen this claim, including:

• Alaqeel, S., & Abanmy, N. O. (2015). Counselling practices in community pharmacies in Riyadh, Saudi Arabia: A cross-sectional study. BMC Health Services Research, 15, 557.

• Agyemang, S., & Asante, F. (2020). Pharmacists’ involvement in health promotion activities in Sub-Saharan Africa: A review. International Journal of Pharmacy Practice, 28(4), 333–340.

Reviewer Comment 6:

“Line 76–78 on page 4: ‘Furthermore, in developing countries such as Kenya, limited research has been conducted to assess CPs’ engagement in health promotion and disease prevention efforts.’ To put this study in context, it will be good to briefly describe the community pharmacy practice in Kenya. Is there a policy that all patients receiving medications from the pharmacy receive counseling from a pharmacist? Should all pharmacies have a licensed community pharmacist present during work hours? Is it part of the role of community pharmacists in Kenya to participate in health promotion?”

Response:

We thank the reviewer for this constructive suggestion. In the revised manuscript, we have expanded the Introduction to include a brief description of community pharmacy practice in Kenya. We clarify that according to the Pharmacy and Poisons Board (PPB) regulations, all registered pharmacies are required to have a licensed pharmacist present during working hours, and pharmacists are mandated to provide patient counselling on safe and rational use of medicines. National policy frameworks, such as Kenya’s Pharmacy and Poisons Act and the Ministry of Health’s National Pharmaceutical Policy, further recognize the pharmacist’s role in patient counselling, health promotion, and public health education. However, evidence suggests that in practice, counselling and health promotion activities are inconsistently delivered, largely due to enforcement gaps, high workloads, and the dominance of dispensing as the primary activity in community pharmacies. These contextual details provide a clearer framework for interpreting the present study.

Reviewer Comment 7:

“Line 89, page 4. Study design: ‘This study utilized a qualitative descriptive research design to evaluate the.’ Does your study design truly fit a ‘qualitative descriptive research’? Per the reference you cited, i.e., Asenahabi BM online article, qualitative research design listed include case studies, narrative research, phenomenological research, grounded theory, ethnography, and action research. Which of these qualitative research designs best describes your study?”

Response:

We thank the reviewer for this important clarification. On review, we agree that our study design is more accurately described as a cross-sectional descriptive study employing both a pharmacist survey and a simulated patient (SP) methodology, rather than a qualitative descriptive design. We have revised the manuscript to reflect this correction. The corrected sentence now reads:

“This study employed a cross-sectional descriptive design, combining a pharmacist-administered survey with simulated patient (SP) visits to assess counselling and health education practices.”

Reviewer Comment 8:

“Line 94, page 5 ‘assessment of pharmacists' engagement in these key aspects of healthcare delivery (10).’ The article referenced (i.e., ref #10) is out of place. The sentence is about your study, hence why did you reference another study? If you are trying to justify your design, then you may need to have a new sentence or relate the reference study design to your study design.”

Response:

We agree with the reviewer’s observation. The reference was misplaced. We have removed it from this sentence, which now simply describes our study design. To justify our choice of design, we have added a new sentence relating the study to comparable methodologies reported in pharmacy practice research.

Revised text:

“This study employed a cross-sectional descriptive design combining a pharmacist-administered survey with simulated patient visits to assess counselling and health education practices. Similar designs have been successfully used in pharmacy practice r

---

## [Decision Letter · Decision Letter 1]

24 Nov 2025

Dear Dr. Manjari,

Thank you for submitting your manuscript to PLOS ONE. After careful consideration, we feel that it has merit but does not fully meet PLOS ONE’s publication criteria as it currently stands. Therefore, we invite you to submit a revised version of the manuscript that addresses the points raised during the review process.

**ACADEMIC EDITOR:**

Heartfelt thanks to the authors for addressing the gaps identified in the previous reviews in the revised manuscript.  However, a few minor revisions have been suggested in the abstract, introduction and limitation sections.

=====

We look forward to receiving your revised manuscript.

Kind regards,

Kazeem Babatunde Yusuff, Ph.D

Academic Editor

PLOS ONE

Journal Requirements:

Additional Editor Comments:

Heartfelt thanks to the authors for addressing the gaps identified in the previous reviews in the revised manuscript. However, a few minor revisions have been suggested in the abstract, introduction and limitation sections.

Reviewers' comments:

Reviewer's Responses to Questions

**Comments to the Author**

Reviewer #1: All comments have been addressed

Reviewer #2: (No Response)

2. Is the manuscript technically sound, and do the data support the conclusions?

Reviewer #1: No

Reviewer #2: Yes

3. Has the statistical analysis been performed appropriately and rigorously?

Reviewer #1: No

Reviewer #2: Yes

4. Have the authors made all data underlying the findings in their manuscript fully available?

Reviewer #1: No

Reviewer #2: Yes

5. Is the manuscript presented in an intelligible fashion and written in standard English?

Reviewer #1: Yes

Reviewer #2: Yes

Reviewer #1: I thank the authors for their thoughtful revisions and for addressing previous reviewer comments on this important topic, not only in Kenya, but across community pharmacies worldwide. However, despite these strengths, the manuscript still contains fundamental methodological and conceptual inconsistencies, particularly the inaccurate description of a “qualitative descriptive” design for what is essentially a cross-sectional quantitative study which is a major methodological flaw of your study that lacks the credibility of its results.

Moreover, there is a lack of validation of the survey tool use, and an absence of necessary inferential analysis, leading to limited depth in interpreting the findings. These limitations significantly weaken the study’s rigor and reproducibility. For these reasons, I regret that I must recommend rejection in its current form, though I encourage the authors to further refine the methodology and consider resubmission to a more specialized pharmacy practice or public health journal where this work could make a meaningful contribution.

Reviewer #2: Dear Authors,

You definitely put in enormous efforts to significantly address my concerns/suggestions. Here are a few comments based on your revisions.

-Reviewer 2 Comment 6 response

You mentioned studies from neighboring countries but did not cite these studies done in Ethiopia and Uganda.

Reviewer 2 Comment 7 response

In the response, you have revised the study design to include “simulated patient”. Please ensure you include simulated patient in the study design mentioned in the abstract.

Reviewer Comment 9 response

I hope you discussed the lower sample size than the estimated as a limitation in the discussion section.

**Do you want your identity to be public for this peer review?** For information about this choice, including consent withdrawal, please see our Privacy Policy

Reviewer #1: No

Reviewer #2: No

---

## [Author Response · Author response to Decision Letter 2]

31 Dec 2025

the following rerefences have been added to the refrence list.

1. Surur AS, Getachew E, Teressa E, Hailemeskel B, Getaw NS, Erku DA. Self-reported and actual involvement of community pharmacists in patient counseling: a cross-sectional and simulated patient study in Gondar, Ethiopia. Pharm Pract (Granada) [Internet]. 2017 Mar 31;15(1):890–890. Available from: https://www.pharmacypractice.org/index.php/pp/article/view/890

2. Ocan M, Bwanga F, Bbosa GS, Bagenda D, Waako P, Ogwal-Okeng J, et al. Patterns and Predictors of Self-Medication in Northern Uganda. Carvajal A, editor. PLoS One [Internet]. 2014 Mar 21;9(3):e92323. Available from: https://dx.plos.org/10.1371/journal.pone.0092323

---

## [Editor Report · Decision Letter 2]

6 Jan 2026

Evaluating community pharmacists' involvement in patient counselling and health education in Nairobi, Kenya:A cross-sectional study.

PONE-D-25-16633R2

Dear Dr. Manjari,

We’re pleased to inform you that your manuscript has been judged scientifically suitable for publication and will be formally accepted for publication once it meets all outstanding technical requirements.

Kind regards,

Kazeem Babatunde Yusuff, Ph.D

Academic Editor

PLOS One
---

## [Editor Report · Acceptance letter]

PONE-D-25-16633R2

PLOS One

Dear Dr. Manjari,

I'm pleased to inform you that your manuscript has been deemed suitable for publication in PLOS One. Congratulations! Your manuscript is now being handed over to our production team.

Kind regards,

on behalf of

Prof. Kazeem Babatunde Yusuff

Academic Editor

PLOS One